∂ | **Open Peer Review** | Parasitology | Research Article

# *Entamoeba histolytica* "mutator" strain with a high rate of genetic mutations assists the elucidation of drug resistance mechanisms

Yumiko Saito-Nakano,[1] Shinji Izumiyama,[1] Makoto Hirai,[2] Tetsuro Kawano-Sugaya,[3] Ghulam Jeelani,[3] Sanjib K. Sardar,[3,4] Sandipan Ganguly,[4] Tomoyoshi Nozaki[3,5]

**ABSTRACT** In drug discovery, target identification and elucidation of resistance mechanisms are essential. The pathogen strains resistant to a compound of interest are useful for these purposes. As the generation of drug-resistant strains is time-consuming and often burdensome, we generated the *Entamoeba histolytica* strain with a high accumulation rate of genetic mutations by introducing proofreading-deficient, error-prone DNA polymerase δ mutant gene under the regulation of a tetracycline-regulatable promoter. We validated this "mutator" strain by showing higher genetic mutations accumulated during *in vitro* passage and, as a proof of concept, by identifying genes and their mutations responsible for resistance against miltefosine. Whole-genome analyses of the mutator strain after 12, 33, and 66 weeks of cultivation in the presence of tetracycline revealed that mutations accumulated in a time-dependent fashion, and the mutation rate of the mutator strain was approximately 60-fold higher than the mock control strain. The highly miltefosine-resistant irreversible clones were isolated from mutator-66 weeks but not from mutator-12 weeks. Whole-genome sequencing analysis of the three miltefosine highly-resistant clones identified shared mutations in three candidate genes potentially responsible for miltefosine resistance. Among them, a mutation in P4-ATPase (EHI_096620$^{N417K}$) was worth noting, as mutations in this gene have previously been implicated in miltefosine resistance in *Leishmania* and *Saccharomyces*. We further demonstrated that exogenous expression of EHI_096620$^{N417K}$ (P4-ATPase) and EHI_035500$^{N182I}$ (kinase) confers miltefosine resistance. The *E. histolytica* mutator is a powerful tool for elucidating resistance mechanisms and potentially the modes of action of existing and future drugs against amebiasis.

**IMPORTANCE** The protozoan parasite *E. histolytica* causes invasive amebiasis that is endemic in developing countries and is characterized by dysentery and liver abscesses. Metronidazole is the first-line therapeutic drug that has been used for a decade, although several adverse effects were well-documented and the risk of resistance was experimentally demonstrated. The development of alternative drugs with different modes of action is a prerequisite for future amebiasis control. To this end, elucidation of the mechanism of action and resistance of potential new antiamebic compounds is important but often challenging. To assist the process, we developed "mutator" with a high genetic mutation rate by exploitation of low-fidelity error-prone DNA polymerase δ. This genome-wide random mutagenesis system demonstrated in this study has many potentials, including rapid identification of mutations associated with resistance against new therapeutic candidates.

**KEYWORDS** *Entamoeba histolytica*, mutator, miltefosine, drug resistance, error-prone DNA polymerase

Address correspondence to Yumiko Saito-Nakano, yumiko@niid.go.jp, or Tomoyoshi Nozaki, nozaki@m.u-tokyo.ac.jp.

The authors declare no conflict of interest.

See the funding table on p. 16.

Antimicrobial resistance is a global public health concern (1–3). It is commonly accepted that all human pathogens, including parasites, have an ability to acquire resistance to existent and new drugs (4–6). Therefore, elucidating the mechanisms of drug resistance is essential to formulate strategies to mitigate the emergence of resistance and to identify new targets for the generation of novel drug candidates. Toward this end, the development of new genetic tools to facilitate rapid elucidation of drug resistance mechanisms and the mode of action of the drug leads is always needed. The generation of drug-resistant parasites often requires time-consuming and labor-intensive steps, including cultivation of wild-type drug-sensitive parasites with the compounds of interest (7–10). Various random mutagenesis techniques, such as chemical mutagens including ethyl methanesulfonate, ultraviolet irradiation, and PCR-based random mutagenesis, have been used to generate resistant mutants (11–13). However, these techniques often cause substantial cell damage. Recently, a novel genome-wide random mutagenesis system using proofreading-deficient DNA polymerase δ (PolDel) has been developed in yeast *Saccharomyces cerevisiae* (14), mice (15), and the malaria parasites *Plasmodium berghei* and *P. falciparum* (16–18). The high fidelity of eukaryotic DNA replication relies on exonucleolytic proofreading and the DNA mismatch repair functions of PolDel (19). Organisms expressing proofreading-deficient PolDel demonstrated a 10- to 100-fold increase in the spontaneous mutation rate (14, 15, 17, 18). These variants are called "mutators" (20). Exploiting this technology, "mutator" mice showed a higher rate of developing cancer than the wild type (15). In addition, *S. cerevisiae* mutator (ScMutator) simplified the isolation of mutant strains (21, 22), and *P. berghei* and *P. falciparum* mutator (PbMutator and PfMutator) enabled the rapid screening of drug-resistant parasites (18, 23).

Although many mutant strains have been isolated from mutators in several organisms, no causative gene responsible for drug resistance has been identified (14, 15, 23), except in the cases of PbMutator and PfMutator. PbMutator allowed the identification of a piperaquine resistance-associated gene (24) and single nucleotide polymorphism (SNP) (23), while PfMutator enabled the identification of KAE609- (cipargamin) and MMV665794- (quinoxaline) resistant SNPs (18). In these studies, it was shown that the base-substitution rate was 36.5-fold and 13- to 28-fold higher in PbMutator and PfMutator than in the wild-type strain, respectively (16–18). These observations provide evidence that mutators could serve as a powerful forward genetic tool.

*Entamoeba histolytica*, the causative agent of amebiasis, is an intestinal protozoan parasite (25), responsible for significant mortality and morbidity in low- and middle-income countries (26). Metronidazole is the most commonly used drug of choice (27, 28). It is referred to as a "prodrug" that needs to be reduced at its nitro group to become effective through the activity of thioredoxin reductase and ferredoxin, transforming it into reactive intermediates, such as nitro radical anion and nitrosoimidazole under anaerobic conditions (6). These toxic metabolites damage the DNA and suppress protein synthesis, which is fatal for the parasite (6). Although clinical metronidazole resistance cases have not been reported in sufficient numbers, *E. histolytica* showed resistance under laboratory conditions (9). Thus, alternative drug targets with new modes of action for treating amebiasis are required (28). A number of compounds have been identified to possess amoeba-specific inhibitory activities, but none of them have been further developed for human use (29–34). Understanding the resistance mechanisms and the mode of action of new drug candidates with genetically modified "mutator" that allow rapid isolation of resistant strains is urgently needed (7, 10). To this end, we generated an *E. histolytica* mutator (EhMutator) by exogenously exploiting proofreading-deficient *E. histolytica* DNA polymerase δ (EhPolDel) mutant. We provided a proof of concept of this new genetic tool by elucidating the resistance mechanism against an experimental compound, miltefosine. The genome of *Entamoeba* is aneuploid and often exceeding tetraploid in size (35). This complexity has hindered the establishment of genome editing for this organism. However, the EhMutator significantly accelerates the understanding of

resistance mechanisms and could become the valuable method for genetic engineering of this parasite in the near future.

## RESULTS

### Generation of an *E. histolytica* strain that expresses proofreading-deficient mutant DNA polymerase δ

*E. histolytica* possesses a single-copy gene encoding the ortholog of DNA polymerase δ (PolDel) catalytic subunit (EHI_006690). This gene is located on contig DS571197 in GenBank and has recently been mapped to chromosome 1 (chr1, position 695661-698894) in the genome, within a tetraploid region. The encoded protein shares overall amino acid identities of 46% and 43% with *S. cerevisiae* Pol3p (11) and *P. berghei* PbPolDel (PBANKA_0501300) (16), respectively. To exogenously express the proofreading-deficient *E. histolytica* PolDel (EhPolDel), the residues D263 and E265, corresponding to orthologous residues in other organisms (19, 22) and known to be essential for proofreading activity, were replaced with alanine (EhPolDel$^{AA}$). Additionally, a FLAG tag was inserted immediately after the nuclear localization signal (NLS) (Fig. S1). The fusion construct NLS-FLAG-EhPolDel$^{AA}$ was inserted into a tetracycline-inducible vector pEhTex/HA to generate the pNLS-FLAG-EhPolDel$^{AA}$ plasmid (Fig. 1A). The mock vector (pEhTex/HA, referred to as pMock) or the pNLS-FLAG-EhPolDel$^{AA}$ plasmid was introduced into HM-1:IMSS clone 6 (HM-1), and transformant cells were selected using G418. The expression level of EhPolDel mRNA was increased more than 12-fold upon tetracycline addition (Fig. S2A). Immunoblot analysis detected a single 124 kDa band corresponding to full-length EhPolDel in a tetracycline-dependent manner (Fig. S2B). The EhPolDel mutant expression induced by tetracycline did not affect cell growth (Fig. S2C). Upon tetracycline induction, cells were subjected to fractionation after mechanical homogenization. An apparently homogeneous 124 kDa band in the membrane (p13) and the cytosolic (s100) fractions (Fig. 1B) indicated that NLS-FLAG-EhPolDel$^{AA}$ was localized in the heavy particulate (p13, such as nuclei and ER) and cytosol (s100). We further examined intracellular localization of NLS-FLAG-EhPolDel$^{AA}$ by immunofluorescence assay (Fig. 1C). All trophozoites showed significant signal in the nuclei, as well as partially localized in the cytoplasm. The position of the FLAG tag was found critical for nuclear localization of EhPolDel$^{AA}$, given that NLS-EhPolDel$^{AA}$-FLAG (FLAG tag fused at the C-terminus, Fig. S3A) was localized in the cytoplasm (Fig. S3B). Taken together, the exogenously expressed NLS-FLAG-EhPolDel$^{AA}$ was expected to play a designated role.

### Enhancement of genetic mutations in EhMutator, demonstrated by whole-genome sequencing analysis

Transformant strains carrying either the pNLS-FLAG-EhPolDel$^{AA}$ or pMock plasmid (Fig. 1) were maintained in the presence of tetracycline for up to 66 weeks to ensure continuous induction of NLS-FLAG-EhPolDel$^{AA}$ (Fig. 2A). Whole-genome sequencing of "mutator" and mock control strains was conducted after the cultivation of 12, 33, and 66 weeks, which yielded 80M ~ of reads (Table S1). Since the *Entamoeba* genome is principally tetraploid (35), in this study, SNPs are defined as nucleotide substitutions with an allele frequency of ≥20% in the regions with ≥50-fold coverage (Fig. S4A). To exclude preexisting SNPs in the parental strain, HM-1 (35), SNPs shared in the EhMutator strains (EhMutator-12w, EhMutator-33w, and EhMutator-66w) were filtered by subtracting SNPs found in mock strains (mock-12w, mock-33w, mock-66w) across the cultivation period (Fig. S5). Multiple SNPs were identified in both mock (Table S2) and EhMutator (Table S3) strains, with the rate increasing in parallel to the tetracycline induction period (Fig. S4B and C). Specifically, only one SNP was identified in both mock-12w and mock-66w strains (Fig. 2B; Table S2). In contrast, 24, 35, and 62 period-specific SNPs were detected in EhMutator-12w, EhMutator-33w, and EhMutator-66w (Fig. 2B). Thus, a total of 121 SNPs were found in EhMutator-66w (Fig. 2B; Table S3). The base-substitution mutation rate in the EhMutator-66w and mock-66w strains was 0.261 and 0.0043 SNPs per day (121 and 2 SNPs per 66

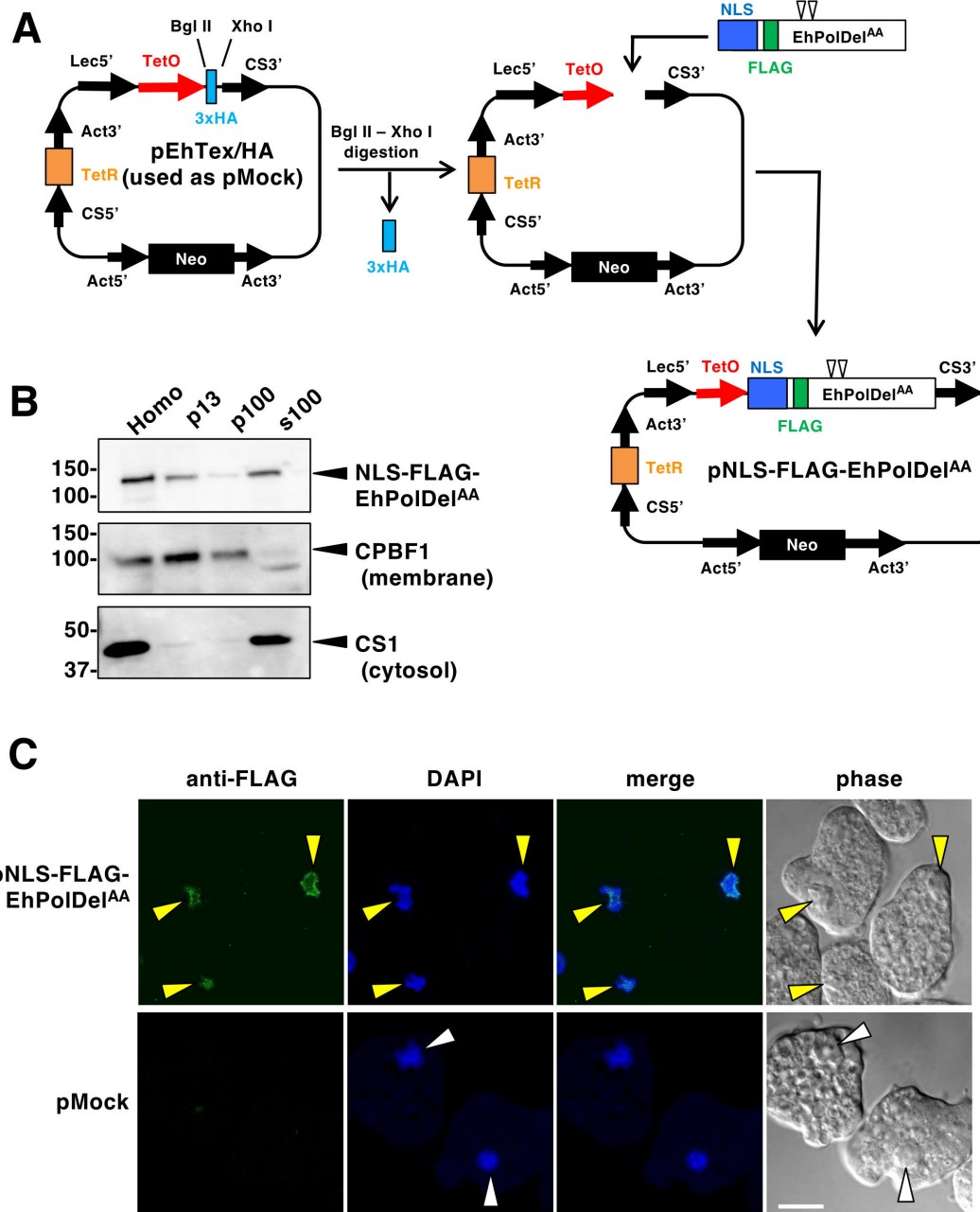

**FIG 1** Nuclear localization of episomally expressed proofreading-deficient DNA polymerase δ mutant. (A) Schematic diagram of the plasmid construction for the generation of *E. histolytica* transformant strain that episomally expressed NLS-FLAG-EhPolDel$^{AA}$. Two mutations resulting in amino acid substitutions, D263A and E265A, are indicated with arrowheads. The position of the FLAG tag was shown in a green box. The tetracycline-inducible vector (pEhTex/HA) was used as mock plasmid (pMock). Abbreviations: tetO, tetracycline-operator; NLS, nuclear localization signal; Lec5′, lectin 5′ untranslated region; CS3′, cysteine synthase 3′ untranslated region; TetR, tetracycline-repressor; Act5′, actin 5′ untranslated region; Act3′, actin 3′ untranslated region; Neo, neomycin resistance gene. (B) Subcellular fractionation of NLS-FLAG-EhPolDel$^{AA}$. Trophozoites were harvested after cultivation with 10 µg/mL tetracycline for 18 hours. Homogenate prepared from transformant cells was fractionated by two-step centrifugation into low-speed pellet (p13, the pellet fraction of 13,000 × g centrifugation), high-speed pellet (p100, the pellet fraction of 100,000 × g centrifugation), and the supernatant fractions (s100, the supernatant fraction of 100,000 × g centrifugation). These fractions were subjected to immunoblot analysis using anti-FLAG, anti-CPBF1 (a transmembrane protein), and anti-CS1 (a cytosolic protein) antibodies. (C) Nuclear localization of NLS-FLAG-EhPolDel$^{AA}$. Trophozoites expressing NLS-FLAG-EhPolDel$^{AA}$ or mock control cells were fixed and stained with anti-FLAG antibodies. Nuclear DNA was stained with DAPI. Yellow arrowheads show colocalization of FLAG and DAPI in the nuclei. Bar, 10 µm.

weeks), respectively, indicating that the mutation rate in EhMutator-66w was 60-fold higher than the mock-66w. Out of the mutations, 45 and 76 SNPs were detected in the

intergenic and coding regions, respectively (Table S3). Of the 76 SNPs in the coding region, seven SNPs resulted in synonymous substitutions, and 69 SNPs resulted in non-synonymous substitutions (Table S3). No insertion or deletion was detected (Table S3). The number of mutations per contig after normalization based on length was comparable among contigs, suggesting that no mutation-prone regions are present in the genome (Table S4). The mutation spectra were transversion-dominant: 95% of the nucleotide substitutions were transversions (36 A:T to C:G substitutions and 79 A:T to T:A substitutions), while only 5% were transitions (6 A:T to G:C substitutions). All mutations occurred in the A and T bases, and none were detected in G and C (Fig. 2C).

## Proof of concept for the use of EhMutator to elucidate the mechanism of drug resistance

To evaluate the value of EhMutator as a tool for the elucidation of the mechanisms of drug resistance, we chose miltefosine (hexadecylphosphocholine) as an anti-amebic compound. *In vitro* amebicidal activity of miltefosine was previously demonstrated with the $IC_{50}$ values of 15–21 µM (36, 37). We first examined the *in vivo* efficacy of miltefosine using a hamster liver abscess model (Fig. 3A). Subcutaneous administration of miltefosine reduced the abscesses to 26.9 ± 8.6% of the total liver weight (41.2 ± 4.8% in untreated hamsters), showing modest efficacy of miltefosine via the subcutaneous route (Fig. 3B). As a positive control, oral administration of metronidazole effectively reduced abscesses (7.6% ± 4.5%) (Fig. 3B).

To understand the mechanism of miltefosine resistance in *E. histolytica*, miltefosine-resistant strains were selected by cultivating EhMutator in the ascending concentrations of miltefosine, from 13 µM to 80 µM, in the absence of tetracycline and G418 to prevent additional mutations (Fig. S6A). The miltefosine-resistant strains (MilR-66w and MilR-79w) emerged earlier than MilR-12w, which was isolated after 42 days, whereas MilR-66w and MilR-79w appeared within 14 days (Fig. S6B). The three resistant strains, MilR-12w, MilR-66w, and MilR-79w (Fig. S6A), were unable to grow in the presence of 6 µg/mL G418 (data not shown) and tested negative for the neomycin resistance gene (Fig. S6C), confirming the loss of the pNLS-FLAG-EhPolDel$^{AA}$ plasmid during miltefosine selection (without G418 and tetracycline). Subsequently, miltefosine-resistant clones were obtained by limiting dilution of three miltefosine-resistant strains. One clone each was obtained from MilR-12w (MilR-12w-clone 3) and MilR-66w (MilR-66w-clone 3), while two clones were from MilR-79w (MilR-79w-clone 1 and 79w-clone 2) (Fig. 4A). The $IC_{50}$ values of miltefosine for MilR-12w-clone 3 and MilR-66w-clone 3 were 4.8- and 8.3-fold higher, respectively, than that of the wild-type HM-1 (Fig. 4B). The $IC_{50}$ values of MilR-79w-clone 1 and MilR-79w-clone 2 were similar to that of MilR-66w-clone 3 (Fig. 4B and C). The stability of the resistant phenotype was investigated by culturing without miltefosine for nine weeks (Fig. 4C). MilR-12w-clone 3 reverted, and its $IC_{50}$ of miltefosine decreased by 63% (77.3 ± 11.4 µM, $P < 0.05$) after 9 weeks. In contrast, the $IC_{50}$ values of MilR-66w-clone 3, MilR-79w-clone 1, and MilR-79w-clone 2 did not alter markedly after cultivation without miltefosine (Fig. 4C). This result suggests that the resistant phenotype of these miltefosine-resistant clones was stable.

## Whole-genome sequencing of miltefosine-resistant clones identified P4-ATPase and kinase being associated with miltefosine resistance

The SNPs that are associated with miltefosine resistance were extracted from the irreversible miltefosine-resistant clones, MilR-66w-clone 3, 79w-clone 1, and 79w-clone 2 (Fig. S7), and commonly shared SNPs are summarized (Table S5). Since mutations in the protein coding regions are of primary interest, we extracted SNPs from the coding regions of genes conserved across *Entamoeba* species (*E. histolytica*, *E. moshkovskii*, *E. nuttalli*, and *E. invadens*). Only those SNPs that generated non-synonymous changes with a frequency of >30% were considered (Table S5). Among 14 SNPs identified, three candidate genes, EHI_096620 (33% identity and e-value $1 \times 10^{-174}$ to human P4-ATPase1A), EHI_035500 (45% identity and e-value $5 \times 10^{-106}$ to human SPRK1 kinase),

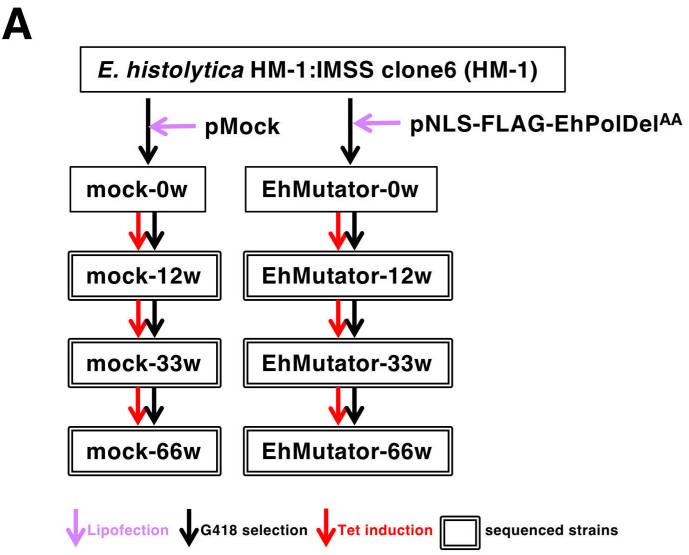

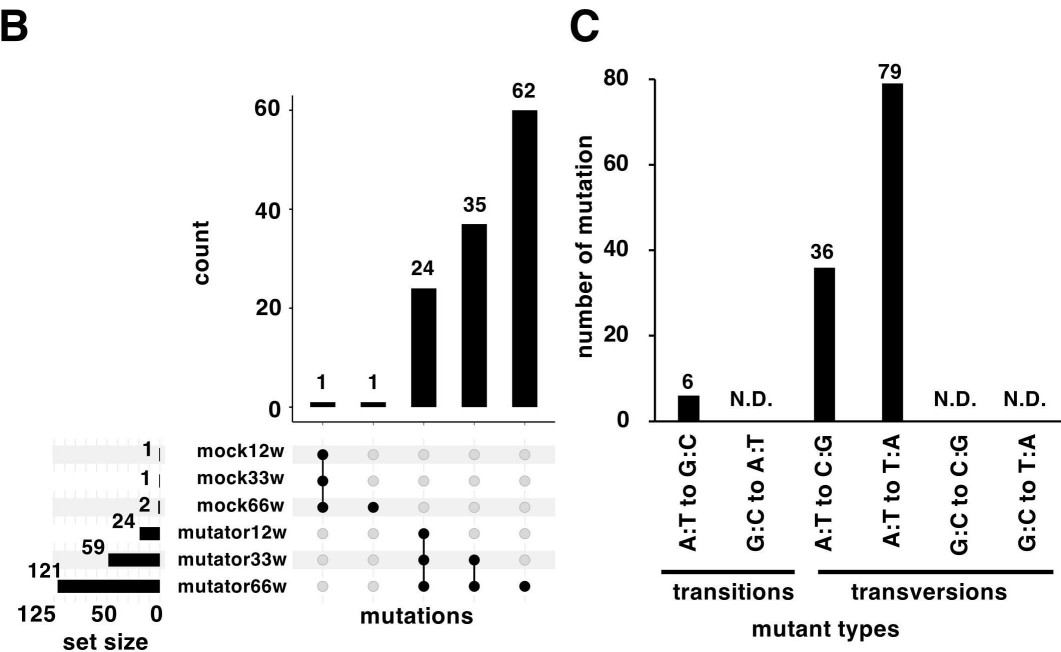

**FIG 2** Establishment of EhMutator and mock strains. (A) The scheme for a cascade of the generation of mutator and mock strains through introduction of plasmids (violet arrows). Both strains were continuously maintained under 6 µg/mL G418 (black arrows), and NLS-FLAG-EhPolDel^AA expression was induced with 10 µg/mL tetracycline (red arrows) for 12, 33, or 66 weeks. Strains subjected to genomic sequencing were surrounded by doublet lines. (B) UpSet plot showing the time-dependent accumulation of SNPs in EhMutator strain. In EhMutator-12w, 24 SNPs were identified and subsequently inherited by EhMutator-33w and EhMutator-66w. Additionally, 35 new SNPs that emerged in EhMutator-33w were also present in EhMutator-66w. Furthermore, 62 additional SNPs were newly detected in EhMutator-66w. Only two SNPs were commonly detected in mock strains. (C) Mutational spectra of the EhMutator-66w strain. Complementary mutations, such as A → C and T → G, are combined.

and EHI_008150 (transmembrane protein), were selected for downstream analyses (Table 1). The first candidate, EHI_096620, showed homozygous substitution, while the other two candidates, EHI_035500 and EHI_008150, showed heterozygosity (Table 1). Asp417 in EHI_096620 is also conserved among *S. cerevisiae*, humans, and *L. donovani* (Fig. S8). Asn 182 in EHI_035500 and Leu 20 in EHI_008150 are also conserved among the other *Entamoeba* spp. (Fig. S9).

To verify the identified mutations are responsible for miltefosine resistance, the parental HM-1 strain was transformed to allow expression of either EHI_096620[N417K] or EHI_035500[N182I] under the tetracycline-inducible system (Fig. S10A). The establishment of a transformant expressing EHI_008150[L20F] failed despite repeated attempts. Transformants expressing EHI_096620[N417K] or EHI_035500[N182I] showed slight miltefosine resistance when expression was induced with 1 µg/mL tetracycline ($P = 0.01$ and $0.02$ vs mock, respectively) (Fig. 5A). EHI_035500[N182I] expressing strain showed weak resistance in the absence of tetracycline ($P = 0.03$ vs mock), which may suggest leaky expression even in the absence of tetracycline. EHI_096620[N417K] or EHI_035500[N182I]-expressing transformants showed miltefosine sensitivity comparable to the mock control strain when expression was induced with 10 µg/mL tetracycline (Fig. 5A). These results indicate that an adequate level of expression of EHI_096620[N417K] or EHI_035500[N182I] confers moderate miltefosine resistance.

For further demonstration of resistance by the reverse method, the wild-type EHI_096620[WT] and EHI_008150[WT] were introduced into MilR-66w-clone 3 using the same

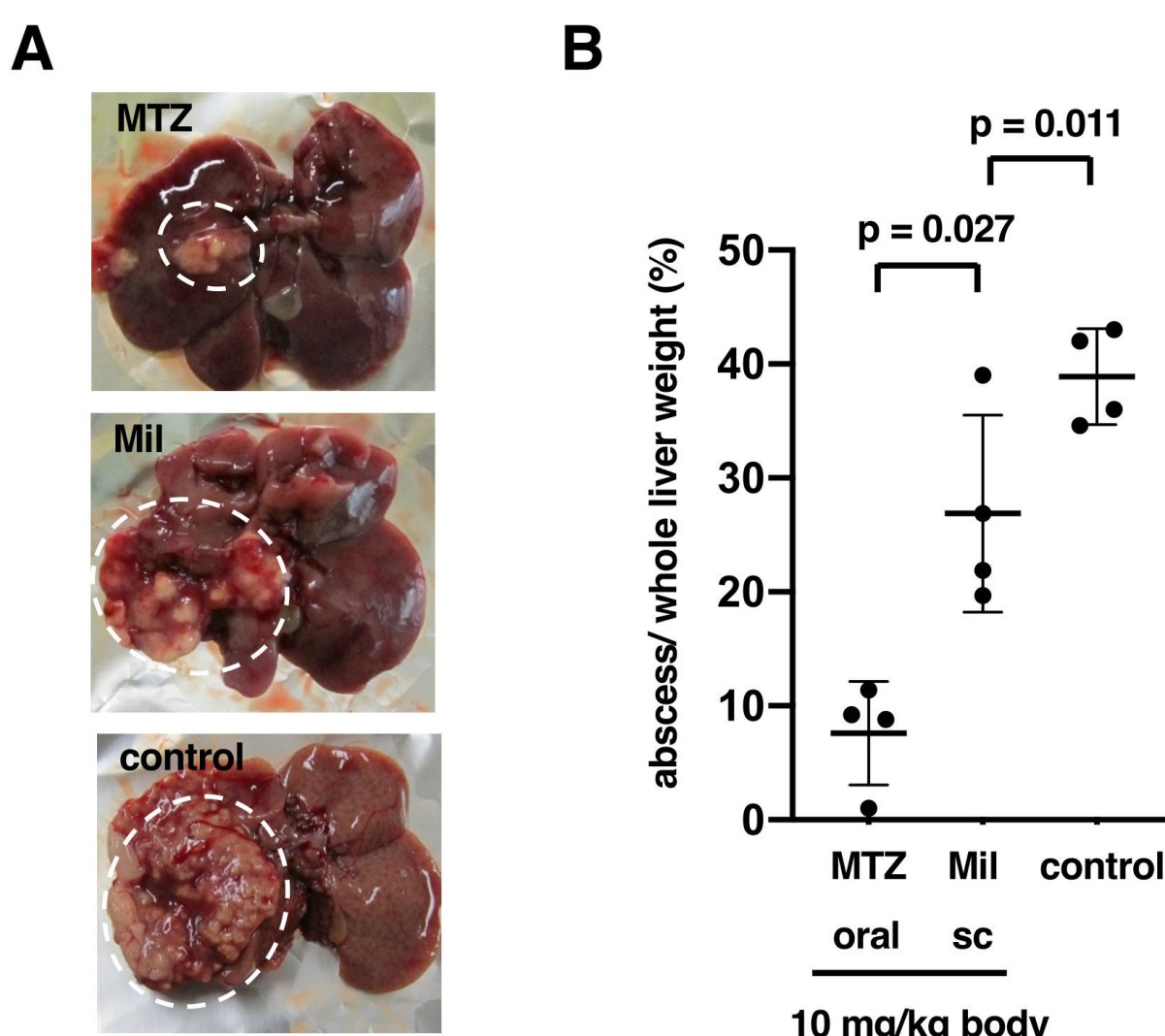

**FIG 3** Efficacy of miltefosine against amebic liver abscesses. (A) The virulent HM-1:IMSS trophozoites were surgically injected into the hamster liver. Metronidazole (MTZ) or miltefosine (Mil) was given at a daily dose of 10 mg/kg body orally or subcutaneously for five days, respectively. Control hamsters were given water orally. Representative image of liver abscesses in hamsters (white dotted circles) with or without drug administration. (B) Evaluation of the efficacy of miltefosine against amebic liver abscesses. The percentage of the weight of amebic liver abscesses per whole liver. After a successive once-a-day drug administration for five days, the weight of the liver abscesses and the whole liver was measured ($n = 4$). Statistical significance was determined using Student's $t$-test.

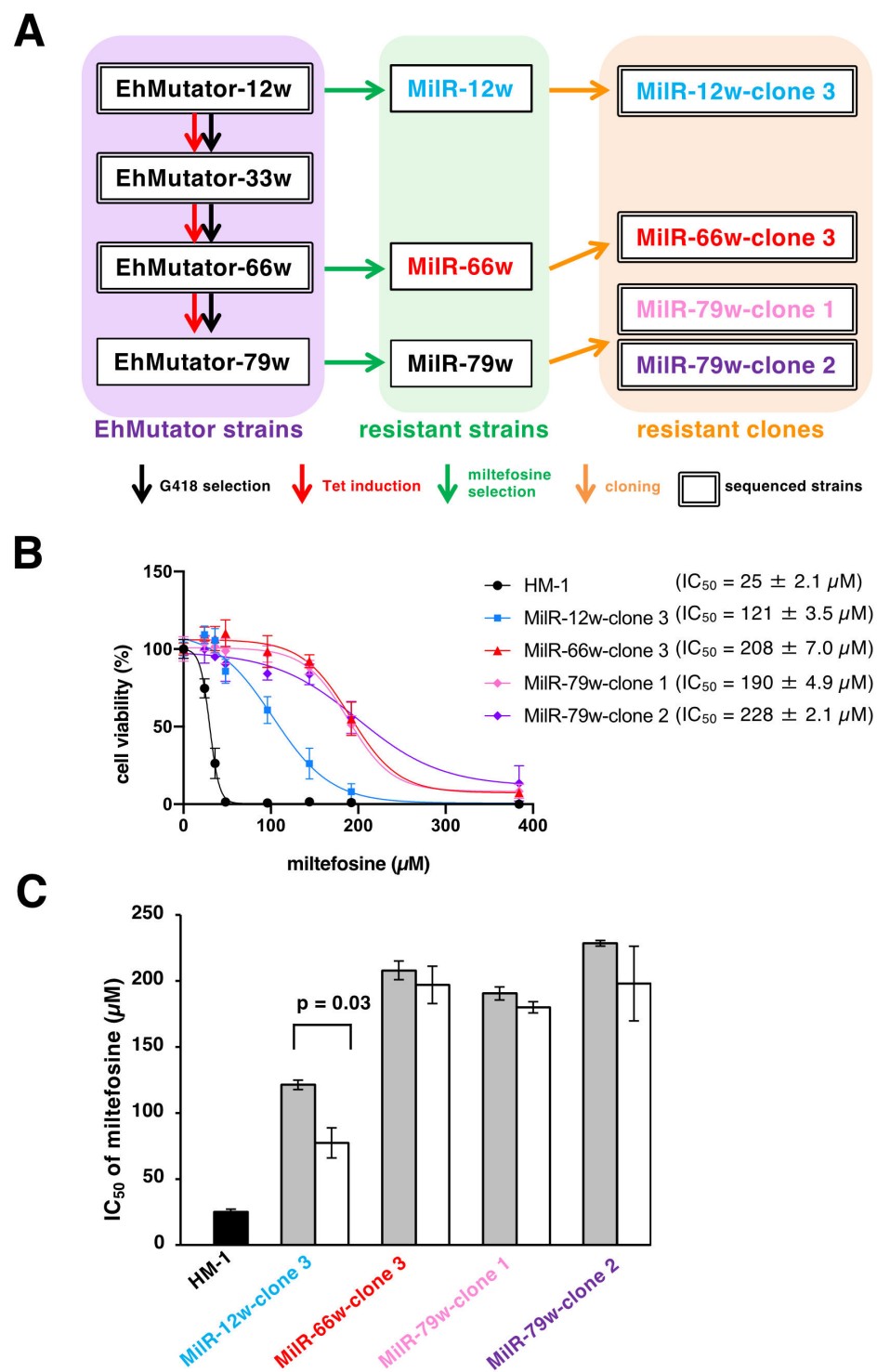

**FIG 4** Generation of miltefosine-resistant *E. histolytica* strains. (A) A schematic diagram of a cascade for the generation of miltefosine-resistant strains and clones. EhMutator strain was cultivated with G418 (black arrows) and tetracycline (red arrows) to allow expression of EhPolDel[AA] for 12, 66, and 79 weeks (EhMutator-12w, EhMutator-66w, and EhMutator-79w) (a purple rectangle), and then subjected to exposure to miltefosine (green arrows) at the step-wise increased concentrations. Note that neither tetracycline nor G418 was used during miltefosine resistance generation. Miltefosine-resistant strain obtained under 24 µM miltefosine after 12 weeks was designated as MilR-12w, while the resistant strain obtained under 120 µM miltefosine at 66 or 79 weeks post-initiation was designated as MilR-66w or MilR-79w, respectively. One clone from each miltefosine-resistant MilR-12w and MilR-66w (MilR-12w-clone 3 and MilR-66w-clone 3) and two clones from MilR-79w (MilR-79w-clone 1 and

**Fig 4 (Continued)**

MilR-79w-clone 2) (an orange rectangle) were obtained by a limiting dilution (orange arrows). (B) Dose response of survival against miltefosine of wild-type HM1, resistant clones obtained from EhMutator-12w (MilR-12w-clone 3), and three highly resistant clones obtained from EhMutator-66w and EhMutator-79w (MilR-66w-clone 3, MilR-79w-clone 1, and MilR-79w-clone 2). The values are shown in means ± SD of three independent experiments. The $IC_{50}$ values represented as means ± SD were shown in parentheses. (C) Resistant phenotypes were irreversible. Miltefosine-resistant clones were maintained with (grey bars) or without (white bars) 24 µM (for MilR-12w-clone 3) or 120 µM (for MilR-66w-clone 3, MilR-79w-clone 1, and MilR-79w-clone 2) miltefosine for nine weeks and then subjected to the miltefosine $IC_{50}$ measurement. The $IC_{50}$ values are the mean ± SD of three independent experiments. The statistical significance was determined using the Student's *t*-test.

tetracycline-inducible system (Fig. S10B). The transformant trophozoite expressing EHI_035500$^{WT}$ failed to establish despite the repeated lipofection. Miltefosine-resistant MilR-66w-clone 3 trophozoites expressing EHI_096620$^{WT}$ were as susceptible to miltefosine as HM-1 under 1 or 10 µg/mL tetracycline ($P = 0.002$ vs mock), and without tetracycline ($P = 0.01$ vs mock) (Fig. 5B). On the contrary, transformants expressing EHI_008150$^{WT}$ did not demonstrate any altered miltefosine sensitivity (Fig. 5B), suggesting that overexpression of EHI_008150$^{WT}$ does not confer miltefosine resistance.

## DISCUSSION

### EhMutator is the powerful tool for forward genetics

This study provides proof of concept for the utility of the mutator technology in identifying drug resistance-associated genes, enabling the rapid isolation of resistant strains in *E. histolytica* for the first time. The implementation of this technology in the research and development of anti-amebiasis drugs is crucial for the understanding of the drug resistance mechanisms and the mode of actions of existing and new drugs against this organism. EhMutator was created by expression of the proofreading-deficient EhPolDel (Fig. 1), which has a 60-fold higher mutation rate than that of the parental strain (Fig. 2B; Table S3). In this study, we have demonstrated that EhMutator allowed the generation of miltefosine-resistant strains and the identification of the genes and mutation sites that are responsible for resistance. P4-ATPase$^{N417K}$ (EHI_096620$^{N417K}$), which we identified as one of the miltefosine resistance genotypes, proved to be miltefosine resistant by reverse genetics (Fig. 5). The P4-ATPase gene was also reported to be responsible for miltefosine resistance in other species (38–40), suggestive of common mechanisms.

The "mutator" technology has been successfully employed in haploid stages of various organisms, including *Plasmodium* and *S. cerevisiae*, in which the intrinsic wild-type allele was replaced with the proofreading-deficient PolDel gene (11, 16, 18). To implement the mutator technology in *E. histolytica*, which has a poly (or aneu)ploid genome, mostly tetraploidy, three technical modifications are required in this study. The first issue was related to construct design—more specifically, the region of epitope

**TABLE 1** Candidate genes and mutations associated with miltefosine resistance

| Gene ID | Contig/chloromosome locus | Ploidy | Amino acid substitution | Zygosity | Allele frequency | | | Product | Comment[a] |
|---------|---------------------------|--------|-------------------------|----------|------------------|--|--|---------|------------|
| | | | | | 66w-clone 3 | 79w-clone 1 | 79w-clone 2 | | |
| EHI_096620 | DS571163/chr6, 615942-619442 | Tetraploid | N417K | Homozygous | 100 | 100 | 100 | P4-ATPase | N417 is conserved in human, *Saccharomyces*, and *Leishmania* |
| EHI_035500 | DS571174/chr9, 1163018-1164133 | Tetraploid | N182I | Heterozygous | 41 | 35 | 38 | SPRK kinase | N182 is conserved in Ei and En |
| EHI_008150 | DS571178/chr4, 306735-307733 | Tetraploid | L20F | Heterozygous | 38 | 37 | 35 | transmembrane protein | L20 is conserved in Em and En |

[a]Ei, *Entamoeba invadens*; Em, *Entamoeba moshkovskii*; En, *Entamoeba nuttalli*.

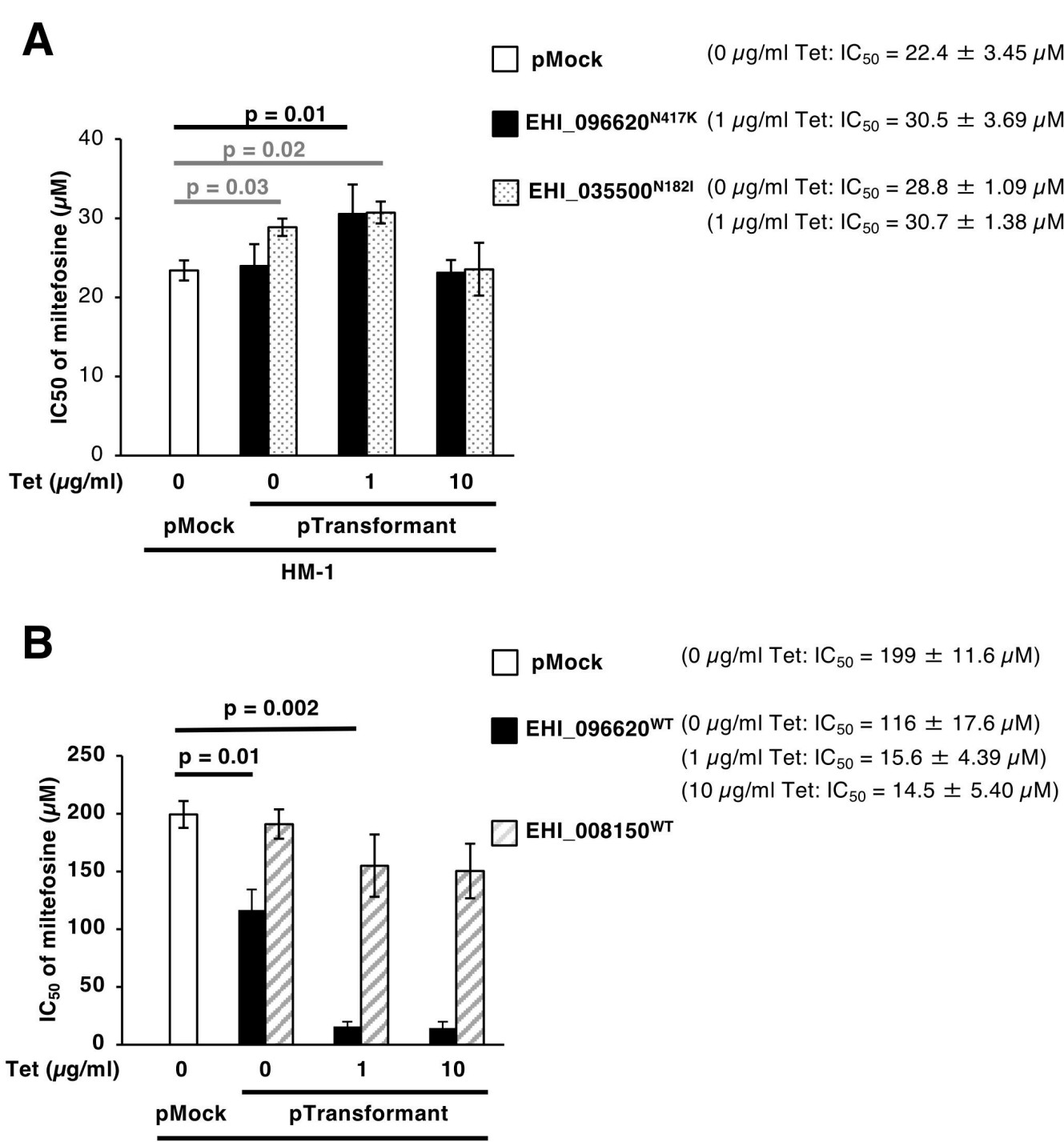

**FIG 5** Validation of EHI_096620 (P4-ATPase)$^{N417K}$ and EHI_035500 (kinase)$^{N182I}$ as miltefosine-resistance-responsible genetic mutations. (A) The IC$_{50}$ value of miltefosine against the transgenic strain where EHI_096620$^{N417K}$ or EHI_035500$^{N182I}$ was expressed in HM-1 strain. Expression was induced under the regulation of the tetracycline (Tet) promoter at the indicated concentrations for 18 hours. The miltefosine IC$_{50}$ values were measured under 0, 1, or 10 μg/mL tetracycline for EHI_096620$^{N417K}$ expressing strain (black bars), EHI_035500$^{N182I}$ expressing strain (dotted bars), and mock control (a white bar). (B) The IC$_{50}$ value of miltefosine against the transgenic strain, where EHI_096620$^{WT}$ or EHI_008150$^{WT}$ was expressed in MilR-66w-clone 3. Expression was induced under the regulation of the tetracycline promoter at the indicated concentrations for 18 hours. The miltefosine IC$_{50}$ values were measured under 0, 1, or 10 μg/mL tetracycline for EHI_096620$^{WT}$ (black bars), EHI_008150$^{WT}$ (hatched bars), and mock control (a white bar). Values are the mean ± SD of triplicate experiments. The IC$_{50}$ values that were statistically significant for the mock are shown in parentheses. The statistical significance was determined using Student's $t$-test.

tagging in EhPolDel[AA]. In *E. histolytica,* the proofreading-deficient EhPolDel[AA] needed to be expressed using an episomal plasmid (Fig. 1), as neither conventional homologous recombination nor genome editing has been established in *E. histolytica*. EhPolDel[AA], when expressed as a fusion protein with the FLAG tag at the C-terminus, showed cytosolic mislocalization (Fig. S3), suggesting that the C-terminus of EhPolDel is likely involved in and necessary for nuclear localization. The importance of the C-terminal domain was previously shown for yeast PolDel (Pol3p), in which the C-terminal CysB element contains a 4Fe-4S cluster and is involved in an interaction with two regulatory subunits during genome replication (41). We therefore added the FLAG tag just after the NLS (Fig. S1), and this protein, NLS-FLAG-EhPolDel[AA], was shown to be localized to the nuclei (Fig. 1C). The second issue was the expression of the PolDel mutant, which was regulated by an inducible promoter derived from an episomal plasmid. In *Plasmodium* and *Saccharomyces*, error-prone PolDel mutants were integrated into the genomic locus through homologous recombination, a mechanism available in these organisms (18, 21– 23). However, mutant clones of *Plasmodium* and *Saccharomyces* isolated from mutators continue to accumulate SNPs unless the PolDel[AA] mutations revert to the wild type. Our episomal system resolved this issue. This system allows for controlled expression of EhPolDel[AA] and enables plasmid loss in the absence of G418 selection. As a result, miltefosine-resistant clones did not express EhPolDel[AA] after a prolonged culture without G418 (Fig. S6). Under the conditions, miltefosine-resistant clones did not revert to susceptible state (Fig. 4C). These findings indicate that, despite the presence of chromosomal copy of wild-type PolDel, the episomal EhPolDel[AA] mutant dominantly influenced the phenotype. The third issue was related to bioinformatics: the criteria (or definition) for identifying SNPs by whole-genome sequencing of aneuploid genomes. As the *Entamoeba* genome is aneuploid, with a majority of regions of chromosomes being tetraploid and some parts being up to septaploid (35), SNPs were analyzed only when substitutions were found with an allelic frequency of ≥20% of the total-read mapping (Fig. S4; Table S3). The number of SNPs in the EhMutator genome increased over time in proportion to the period of parasite proliferation under the tetracycline induction (Fig. S4B and C; Table S3). The SNPs acquired in the early period (12 and 33 weeks) were inherited in the later period (66 weeks) without being lost (Fig. 2B). These data suggest that the SNPs were efficiently accumulated within the alleles. Thus, the "mutator" can also be applied to other eukaryotic organisms where homologous recombination has not been established, such as *Babesia* (42), or protozoa with a polyploid genome, such as *Naegleria* (43) and *Tetrahymena* (44).

EhMutator shows some common features similar to the mutators from other organisms, as well as unique characteristics. EhMutator-66w exhibited a 60-fold increase in the mutation rate compared to the mock control (Fig. 2B). This trend was similar to that observed in *Plasmodium berghei* mutator, PbMutator, which showed a 36-fold higher mutation rate than the wild-type strain at 122 weeks (17). However, the mutation spectra, such as the transition–transversion ratio, differ among organisms and depend upon environmental conditions. EhMutator-66w apparently exclusively drives A:T to C:G and A:T to T:A transversions, with no G:C being replaced with A:T, as judged from mutations detected after 66-week inductions (Fig. 2C). In contrast, it was shown that PbMutator preferentially gained a C:G to A:T transition. Thus, the PbMutator genome tends to shift toward a higher AT content (17). It has been previously shown that nucleotide insertions and deletions are also introduced in PbMutator. However, neither insertions nor deletions were detected in EhMutator (Table S3), which may indicate the distinct mechanisms of mutations introduced by error-prone DNA polymerase δ in *E. histolytica*. A yeast mutator was reported to accumulate both transitions and transversions, leading to more diverse amino acid substitution patterns (11). Although in this study only EhPolDel[AA] double mutant was used, the introduction of an additional mutation of the amino acid residue involved in replication fidelity in PolDel (45) may broaden the mutation spectra and contribute to a more robust generation of drug-resistant *E. histolytica* strains. SNPs were distributed across all contigs, and no clear deviation to

certain contigs or certain regions of contigs was observed (Table S4), indicating that there are not the mutation-prone "hotspot," or there are not a set of essential genes that remain free of SNPs, in contrast to the *Plasmodium* mutator (46).

## Identification of *Entamoeba*-specific miltefosine resistance genes and mutations

No overlapping SNPs were found between EhMutators (before miltefosine screening, Table S3) and miltefosine-resistant candidate genes (after miltefosine screening, Table S5). These findings suggest that the miltefosine-resistant candidate SNPs were selected from pre-existing SNPs in EhMutator with an allelic frequency of less than 20% (Fig. S4A, black dots). Among the SNPs that were found to be associated with miltefosine resistance, one SNP each of two genes was proven to provide resistance phenotypes: homozygous EHI_096620[N417K] and heterozygous EHI_035500[N182I] mutations (Table 1). To the best of our knowledge, EHI_096620[N417K] is the first homozygous mutation identified in the course of the generation of drug resistance in *E. histolytica*. Although a number of studies were conducted to raise drug resistance *in vitro* by gradually increasing drug concentrations in the medium, and changes in gene expression that were associated with reversible (8) and irreversible (7) resistance were documented (7–10, 27), no single mutations responsible for drug resistance have been reported. In the present study, the expression of EHI_096620[N417K] in the parental HM-1 strain resulted in weak miltefosine resistance (Fig. 5A). Conversely, the expression of EHI_096620[WT] in the miltefosine-resistant clone reduced its resistance (Fig. 5B). These results indicate that the EHI_096620[N417K] mutation is recessive and that an allelic frequency of 100% is likely required for miltefosine resistance. In contrast, EHI_035500[N182I] (kinase) was a heterozygous mutation (Table 1), and expression of EHI_035500[N182I] in the parental HM-1 gave weak miltefosine resistance (Fig. 5A). This mutation is a dominant resistant-associated mutation, when introduced into HM-1 (Fig. 5A), indicating that this kinase is the second resistance-associated gene. These genes and mutations were discovered for the first time in *E. histolytica*. Curiously, transformed trophozoites expressing EHI_035500[WT] failed to establish even in the absence of tetracycline, suggesting that a slight increase in EHI_035500[WT] expression might be toxic to the cells. The weak miltefosine-resistant clone MilR-12w-clone 3 exhibited reversible resistance (Fig. 4C). Although the mutations responsible for resistance in MilR-12w-clone 3 were not identified in this study, the number of SNPs introduced in EhMutator-12w was lower than in EhMutator-66w (Fig. S4C). Therefore, the mutation allele frequencies in MilR-12w-clone 3 were expected to be lower than those in MilR-66w-clone 3. In another polyploid amoeba, *Acanthamoeba*, it has been shown that mutations that occurred in one of the four genome copies can be corrected by wild-type allelic copies through gene conversion (43). Similarly, the mutation responsible for miltefosine resistance in MilR-12w-3 may have reverted to wild type during the nine-week removal of miltefosine (Fig. 4C).

The mechanism of miltefosine resistance in *Leishmania donovani* has been suggested by *in vitro* studies and evidence from field and clinical isolates. In these studies, miltefosine resistance was shown to be associated with a decreased accumulation of miltefosine, suggestive of altered uptake, efflux, and/or metabolism (38–40). In *Leishmania*, the inactivation of a P4-ATPase, named miltefosine transporter (LdMT), and its noncatalytic subunit Cdc50 homolog LdRos3 was required for miltefosine resistance (47). LdMT and LdRos3 are involved in the translocation of phospholipids from the exoplasmic to the cytoplasmic leaflet of the plasma membrane; thus, their inactivation reduces the accumulation of miltefosine within the parasite's cytoplasm (47). Several independent recessive point mutations in LdMT have been reported to be responsible for miltefosine resistance (47). Of them, one mutation was found in the consensus residue of the cytosolic ATPase domain, causing T420N mutation in LdMT, which is proximal to the mutation found in *E. histolytica* homolog EHI_096620 (N417K) (Fig. S8). EHI_096620[N417K] is a novel mutation associated with miltefosine resistance first identified in *E. histolytica*.

We previously reported that the overexpression of the amebic homolog of a noncatalytic subunit of P4-ATPase, Cdc50 (EhCdc50), confers miltefosine resistance to *Entamoeba* (37), as observed in mammalian cells and *Arabidopsis* (48, 49). It was explained that this resistant phenotype is caused by the stoichiometric imbalance between Cdc50 and P4-ATPase on the ER membrane because the heterodimeric complex of Cdc50/P4-ATPase needs to be transported from the ER to the plasma membrane to be functional as a phospholipid transporter (50–52). A lack or a decrease of P4-ATPase on the plasma membrane reduces ATP-dependent internalization of phospholipids as well as miltefosine (48, 49). Altogether, our results are consistent with the premise that EHI_096620 (P4-ATPase), among 12 P4-ATPases encoded in the amebic genome (37), and EhCdc50 are co-transported to the plasma membrane and are involved in phospholipid internalization.

Finally, in this study, we have shown the proof of concept that the "mutator" is a powerful tool to generate drug-resistant strains and to elucidate the mechanism of resistance. Thus, the mutator approach can facilitate drug discovery against amebiasis and other protozoan parasites. Furthermore, "mutator" can be utilized for a broad range of scope, including the identification of genes responsible for *in vitro* and *in vivo* virulence and fitness.

## MATERIALS AND METHODS

### Animal experiment and ethics statement

The animal experiments were approved by the Institutional Animal Care and Use Committee (No. 115141-II) and were conducted at the AAALAC-accredited National Institute of Infectious Diseases, Japan. At the end of the experiments, the animals were euthanized by cervical dislocation under isoflurane anesthesia. The construction of mutator *Entamoeba* was approved by the Safety Committee for Genetic Modification Experiments (No. Ki2-3 and Ki6-141).

### Chemicals

Miltefosine hydrate was purchased from the Tokyo Chemical Industry (Tokyo, Japan), and a 10 mg/mL (24.5 mM) stock solution was prepared in water. The stock was sterilized by filtration, stored at 4°C protected from light, and used within one week. Metronidazole was purchased from Sigma-Aldrich (St. Louis, MO, USA), and a 100 mM stock solution was prepared in DMSO. Tetracycline and geneticin (Invitrogen, Carlsbad, CA, USA) were dissolved in 50% ethanol and water, respectively, to obtain 10 mg/mL solutions.

### *Entamoeba* culture

Trophozoites of the attenuated *E. histolytica* strain HM-1:IMSS clone 6 (HM-1) were axenically cultured at 35.5°C in BI-S-33 medium, as previously described (53, 54). Liver-passed virulent HM-1 (v-HM1) was cultured at 35.5°C in YIMDHAS medium supplemented with live *Crithidia fasciculata* (55).

### Plasmid construction

A plasmid expressing the mutant EhDelPol was constructed as follows. First, a 3,234 bp DNA fragment containing EHI_006690 was PCR-amplified using the *E. histolytica* genomic DNA and the YSN665b and YSN672b oligonucleotides (Table S6). *Entamoeba*-expressing plasmid pEhTex/HA (56) was digested with BglII and XhoI to remove the HA-tag sequence, and the amplified fragment was cloned into the BglII-XhoI sites of pEhTex/HA to yield NLS-EhPolDel/pEhTex. A triple-tandem FLAG sequence was inserted with Ile46 residues by PCR amplification of the NLS-EhPolDel/pEhTex using overlapping oligonucleotides YSN712b and YSN713b, yielding the NLS-FLAG-EhPolDel/pEhTex. The Clontech In-Fusion HD Cloning Kit (Thermo Fisher Scientific, Waltham, MA, USA) was

used to construct all the plasmids. Proofreading-deficient mutations D263A and E265A were constructed by PCR-mediated mutagenesis using the PrimeSTAR Mutagenesis Kit (TaKaRa, Shiga, Japan). Miltefosine-resistance-related genes EHI_096620, EHI_035500, and EHI_008150 were inserted into the BglII site of pEhTex/HA to obtain the C-terminal HA fusion.

## Transformation of *E. histolytica*

The plasmids were introduced into wild-type HM-1 by lipofection method, as previously described (57). Transformant amebae were selected by adding 1 µg/mL geneticin (G418) after 24 hours of transfection, which was gradually increased to 6 µg/mL over two weeks. Then, 10 µg/mL tetracycline was added to induce the mutant DNA polymerase.

## DNA preparation and whole-genome sequencing

Trophozoites of the mutator, mock, and miltefosine-resistant clones were cultured in 100 mm polystyrene tissue culture dishes (30 dishes/clone). Then, 2 µg of genomic DNA was extracted from each strain using the Blood & Cell Culture DNA Maxi Kit (Qiagen, Hilden, Germany), following a previously described procedure (35). The purified genomic DNA samples were sequenced using the Illumina HiSeq X platform (Macrogen, Japan), and 150 bp paired-end reads were analyzed.

## Read mapping and SNP analysis

CLC Genomics Workbench software (Qiagen, Hilden, Germany) was used for quality assessment, trimming, subsampling, mapping, and SNP analysis of each sample. SNP detection was performed using the Basic Variant Detection tool with modified program settings: Relative Read Direction Filter > No. SNP calling was performed using *E. histolytica* HM-1:IMSS (58) as the reference genome (https://amoebadb.org/amoeba/app) with a gene model provided by Dr. Hon (59). Based on this definition, >6,000 SNPs were detected in all strains (step 1, Fig. S5). Such a high number suggests that the parental strain HM-1, maintained for 15 years in our lab (35), has diverged remarkably from the HM-1:IMSS reference strain (58). Since most of the *E. histolytica* genome is tetraploid (35), base substitutions with >20% allele frequency in the mappings were defined as SNPs. Next, SNPs with <50 × coverage were removed (step 1, Fig. S5). SNPs shared among mock strains were excluded from the mutator strains (step 2, Fig. S5). Finally, the residual SNPs specific to the mock or mutator were manually inspected (step 3, Fig. S5). Considering the leaky expression of EhPolDel[AA] even in the absence of tetracycline (Fig. S2A), the pMock strain was used as the control instead of the EhMutator strain untreated with tetracycline in this study.

## Identification of miltefosine-resistance-related loci

More than 7,000 SNPs shared among the three highly resistant clones were selected by comparison with the reference genome of *E. histolytica* HM-1 (Fig. S7). SNPs that were also present in mock strains were excluded, reducing the number to 394. Of these, SNPs detected in the original EhMutator before drug selection were further excluded, leaving 130 SNPs. Finally, 24 non-synonymous SNPs across 130 genes were selected (Fig. S7). Among these, 10 genes whose functions seemed to be irrelevant to miltefosine resistance were excluded to collect 14 candidate genes (Table S5).

## Hamster liver abscesses model

Surgically, injection of ~4 × 10⁵ liver-passed v-HM1 was made into the left lobe of the liver of five-week-old Syrian hamsters (Japan SLC Inc. Shizuoka, Japan). Miltefosine or metronidazole (10 mg/kg body weight) was mixed with the equal volume of 5% polyoxyl 35 castor oil (Tronto Research Chemicals Inc., ON, Canada) and administered subcutaneously after 24 hours. Control animals were provided with water. The animals

were administered for five consecutive days and sacrificed on day 6 post-infection. The whole livers and their abscesses were dissected and weighed separately (30).

## Drug screening for miltefosine-resistant strains from EhMutator

The trophozoites of the mutator strains were maintained for 12, 33, 66, and 79 weeks in the presence of tetracycline under miltefosine-free conditions to establish each strain: EhMutator-12w, EhMutator-33w, EhMutator-66w, and EhMutator-79w. Then, the strains were gradually adapted to 24–120 µM miltefosine over 16 weeks to obtain the MilR-12w, MilR-66w, and MilR-79w strains. The clones MilR-12w-clone 3, MilR-66w-clone 3, MilR-79w-clone 1, and MilR-79w-clone 2 were isolated from each MilR line by limiting dilution in BI-S-33 medium with 24 µM miltefosine, which contained 20% culture supernatant and 80% freshly prepared new medium, and co-cultured with *C. fasciculata* (55). The three clones, MilR-66w-clone 3, MilR-79w-clone 1, and MilR-79w-clone 2, were maintained with or without 120 µM miltefosine for three weeks. After that, each clone was subjected to a miltefosine sensitivity assay.

## Drug sensitivity assay

The miltefosine sensitivity assay was conducted according to a previously reported protocol (37). In brief, trophozoites were seeded into 96-well plates at $0.5 \times 10^4$ cells/well in 280 µL of BI-S-33 medium containing 12–384 µM miltefosine. G418 and tetracycline were also added to the strains expressing miltefosine-resistance-related candidate genes. After 18 hours of culture, the medium was removed, and the trophozoites were incubated with 100 µL Opti-MEM medium containing 10% WST-1 reagent (Roche, Basel, Switzerland) for 20 minutes at 37°C. The viability of the attached trophozoites was estimated by measuring the $OD_{450}$ (37). Data analysis was conducted using GraphPad Prism version 8 (GraphPad Software, San Diego, CA, USA).

### ACKNOWLEDGMENTS

We thank Takashi Makiuchi for providing the pEhTex/HA plasmid, Seiki Kobayashi for his valuable technical support in creating the animal models, Kumiko Nakada-Tsukui for their advice on genome analysis, Eiko Nakasone and Yuko Umeki for their technical assistance, and all the members of our laboratory for their valuable discussions.

This study was conducted using the grants for research on emerging and re-emerging infectious diseases from the Japan Agency for Medical Research and Development (AMED, JP23fk0108683 and JP23fk0108680 to Y.S.-N; JP23fk0108680 to M.H.; JP23fk0108680, JP23jm0110022, and JP233fa627001 to T.N.) and from Grants-in-Aid for Scientific Research (JP22K07050, to Y.S.-N.; JP21H02723, JP21K19372, JP21KK135, and JP21H04779 to T.N.) from the Japan Society for the Promotion of Science (JSPS).

### AUTHOR AFFILIATIONS

[1]Department of Parasitology, National Institute of Infectious Diseases, Japan Institute for Health Security, Shinjuku, Tokyo, Japan

[2]Department of Tropical Medicine and Parasitology, Faculty of Medicine, Juntendo University, Bunkyo, Tokyo, Japan

[3]Department of Biomedical Chemistry, Graduate School of Medicine, The University of Tokyo, Bunkyo, Tokyo, Japan

[4]Division of Parasitology, ICMR-National Institute for Research in Bacterial Infections, Kolkata, West Bengal, India

[5]Department of Internal Medicine, Faculty of Medicine, Universitas Airlangga, Surabaya, East Java, Indonesia

## AUTHOR ORCIDs

Yumiko Saito-Nakano  http://orcid.org/0000-0002-7132-194X
Makoto Hirai  http://orcid.org/0000-0002-5001-9653
Tomoyoshi Nozaki  http://orcid.org/0000-0003-1354-5133

## FUNDING

| Funder | Grant(s) | Author(s) |
| --- | --- | --- |
| Japan Agency for Medical Research and Development | JP23fk0108683 JP23fk0108680 | Yumiko Saito-Nakano |
| Japan Agency for Medical Research and Development | JP23fk0108680 JP23jm0110022 JP233fa627001 | Tomoyoshi Nozaki |
| Japan Society for the Promotion of Science | JP22K07050 | Yumiko Saito-Nakano |
| Japan Society for the Promotion of Science | JP21H02723 JP21K19372 JP21KK135 JP21H04779 | Tomoyoshi Nozaki |
| Japan Agency for Medical Research and Development | JP23fk0108680 | Makoto Hirai |

## AUTHOR CONTRIBUTIONS

Yumiko Saito-Nakano, Conceptualization, Data curation, Formal analysis, Funding acquisition, Investigation, Methodology, Project administration, Writing – original draft, Writing – review and editing | Shinji Izumiyama, Data curation, Formal analysis, Project administration, Software, Writing – original draft, Writing – review and editing | Makoto Hirai, Conceptualization, Methodology, Software, Writing – review and editing | Tetsuro Kawano-Sugaya, Data curation, Methodology, Software, Writing – review and editing | Ghulam Jeelani, Data curation, Methodology, Validation, Writing – review and editing | Sanjib K. Sardar, Data curation, Methodology, Validation, Writing – review and editing | Sandipan Ganguly, Conceptualization, Data curation, Methodology, Supervision, Validation, Writing – review and editing | Tomoyoshi Nozaki, Conceptualization, Funding acquisition, Methodology, Project administration, Supervision, Validation, Writing – original draft, Writing – review and editing

## DATA AVAILABILITY

All raw genome sequencing data have been deposited into the DNA Data Bank of Japan (DDBJ) under the accession numbers DRR550136—DRR550144.

## ADDITIONAL FILES

The following material is available online.

### Supplemental Material

**Supplemental material (Spectrum01210-25-s0001.pdf).** Supplemental methods; Fig. S1 to S10.
**Table S1 (Spectrum01210-25-s0002.xlsx).** Summary of the HiSeqX sequencing used in this study.
**Table S2 (Spectrum01210-25-s0003.xlsx).** List of SNPs identified from the mock strains.
**Table S3 (Spectrum01210-25-s0004.xlsx).** List of SNPs identified from the mutator strains.
**Table S4 (Spectrum01210-25-s0005.xlsx).** Number of SNPs per contig size.
**Table S5 (Spectrum01210-25-s0006.xlsx).** Candidate genes responsible for miltefosine resistance.
**Table S6 (Spectrum01210-25-s0007.xlsx).** List of oligonucleotides used in this study.

## Open Peer Review

**PEER REVIEW HISTORY (review-history.pdf).** An accounting of the reviewer comments and feedback.

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
