## [Reviewer comments · Microbiology Spectrum]

Microbiology Spectrum

***Entamoeba histolytica* “mutator” strain with a high rate of genetic mutations assists the elucidation of drug resistance mechanisms**

Yumiko Saito-Nakano, Shinji Izumiyama, Makoto Hirai, Tetsuro Kawano-Sugaya, Ghulam Jeelani, Sanjib Sardar, Sandipan Ganguly, and Tomoyoshi Nozaki

Corresponding Author(s): Yumiko Saito-Nakano, Kokuritsu Kansensho Kenkyujo

Review Timeline:

Submission Date:	April 18, 2025
Editorial Decision:	May 7, 2025
Revision Received:	May 8, 2025
Accepted:	May 15, 2025

Editor: Christina Cuomo

Reviewer(s): The reviewers have opted to remain anonymous.

Transaction Report:

DOI: <https://doi.org/10.1128/spectrum.01210-25>

Re: Spectrum01210-25 (*Entamoeba histolytica* "mutator" strain with a high rate of genetic mutations assists the elucidation of drug resistance mechanisms)

Dear Dr. Yumiko Saito-Nakano:

Thank you for submitting your work to Microbiology Spectrum.

I am pleased to inform you that your manuscript has been editorially accepted for publication. However, there are a few additional questions in the submission form that need to be answered before the final decision. Once these are completed, please return your submission so that I can move your paper forward to acceptance.

Below you will find instructions from the Spectrum editorial office.

Revision Guidelines

Sincerely,
Christina Cuomo
Editor
Microbiology Spectrum

Re: Spectrum01210-25R1 (*Entamoeba histolytica* "mutator" strain with a high rate of genetic mutations assists the elucidation of drug resistance mechanisms)

Dear Dr. Yumiko Saito-Nakano:

Your manuscript has been accepted, and I am forwarding it to the ASM production staff for publication. Your paper will first be checked to make sure all elements meet the technical requirements. ASM staff will contact you if anything needs to be revised before copyediting and production can begin. Otherwise, you will be notified when your proofs are ready to be viewed.

Sincerely,
Christina Cuomo
Editor
Microbiology Spectrum